# Vertical Alignment of Liquid Crystals on Comb-Like Renewable Chavicol-Modified Polystyrene

**DOI:** 10.3390/polym13050819

**Published:** 2021-03-07

**Authors:** Jihyeon Moon, Kyutae Seo, Hyo Kang

**Affiliations:** BK-21 Four Graduate Program, Department of Chemical Engineering, Dong-A University, 37 Nakdong Daero 550beon-gil, Saha-gu, Busan 604-714, Korea; 1829469@donga.ac.kr (J.M.); kyutae@donga.ac.kr (K.S.)

**Keywords:** liquid crystal, alignment, phytochemical, renewable, chavicol

## Abstract

This study demonstrates liquid crystal (LC) alignment behaviors on the surface of phytochemical-based and renewable chavicol-modified polystyrene (PCHA#, # = 20, 40, 60, 80, and 100, where # represent the molar content of chavicol moiety in the side group) via polymer modification reactions. Generally, a LC cell fabricated with a polymer film containing a high molar content of the chavicol side group exhibited a vertical LC alignment property. There is a correlation between the vertical alignment of LC molecules and the polar surface energy value of the polymer films. Therefore, vertical LC alignment was observed when the polar surface energy values of these polymer films were smaller than about 1.3 mJ/m^2^, induced by the nonpolar chavicol moiety having long and bulky carbon groups. Aligning stability under harsh conditions such as ultraviolet (UV) irradiation of about 5 J/cm^2^ was observed in the LC cells fabricated from PCHA100 film. Therefore, it was found that the plant-based chavicol-substituted polymer system can produce an eco-friendly and sustainable LC alignment layer for next-generation applications.

## 1. Introduction

Liquid crystal (LC) molecules are beneficial organic materials with an intermediate phase between crystalline solids and isotropic liquids because of their crystal-like ordering and fluidity [1]. LC molecules are readily responsive to external stimuli such as electric, magnetic fields, and surface interaction, contributing to their rheological behaviors and anisotropic physical properties [2,3]. Their interesting features allow them to be used in numerous applications such as electro-optical components, responsive sensors, and biological applications [4,5,6,7,8,9,10,11,12,13,14,15,16,17,18,19]. For example, LC molecules can be used in several electronic products such as color filters [4], smart glasses [5], and display applications [6,7] owing to their susceptibility to electric fields. Moreover, changes in the optical properties of LCs induced by ordering transitions under external stimuli are visible to the naked eye without additional labels or instruments. Therefore, LCs can be employed as simple monitoring sensors under external stimuli and environments such as temperature [8], gases [9], humidity [10], and indoor residential dust [11]. LC molecules are also studied for utilization as biosensors to detect the presence of proteins [12,13], surfactants [14,15], lipids [16,17], bacteria and virus [18,19]. Control of the alignment behaviors of LC molecules is essential in diverse scientific and technical fields [4,5,6,7,8,9,10,11,12,13,14,15,16,17,18,19,20,21,22,23]. LC alignment methods related to controlling the aligning abilities of LC molecules, such as pretilt angle and anchoring strength, have received attention by other researchers [24,25,26,27]. For example, contact methods such as a rubbing process use aromatic polymers having a rigid backbone such as polyimide derivatives. Most widely used as conventional LC alignment layers because they exhibit strong interaction through π–π and dipole–dipole interactions between polymers and LC molecules and are suitable for providing stable LC alignment [28,29,30,31,32,33,34,35,36,37]. Moreover, polyimide derivatives having long alkyl side groups have been developed as LC alignment layers [38,39,40,41]. However, reliable polyimide films are commonly produced by baking processes using high temperatures over 200 °C, but this is too high to be practical for several applications. Therefore, long alkyl or fluoroalkyl group modified polystyrene derivatives have been developed as an alternative to polyimide derivatives to produce LC alignment layers without the need for baking processes [42]. The long alkyl or fluoroalkyl groups on polystyrene layers with good solubility in many medium-polarity solvents using low boiling points can produce low surface energy values because of the steric effect from the alkyl or fluoroalkyl groups on the surface of polymer films [42].

Chavicol has phenol derivatives with a phenylpropene structure and is extracted from various plants such as *Piper betle* [43,44]. *Piper betle* essential oil containing chavicol plays a vital role in antifungal, antibacterial, and antioxidant activities, preventing various diseases [44,45,46,47,48]. For example, chavicol extracted from *Piper betle* exhibited effective inhibition of bacterial growth against Gram-positive bacteria such as *Staphylococcus aureus* in conjunctivitis patients [45]. The redox properties of phenolic compounds extracted from natural product can exhibit antioxidant activities by the neutralization or binding of free radicals, restraint of singlet and triplet oxygen, and decomposition of peroxides, as previously reported by researchers [48,49,50,51,52]. Furthermore, phenylpropene structure of chavicol has a floral fragrance and can provide specific signals to pollinators. Therefore, essential oil of chavicol has been used extensively in the cosmetics industry, including as an odorant in perfumes, soaps, and food flavorings [53,54].

In this study, comb-like plant-based chavicol-substituted polystyrene (PCHA#) was synthesized to obtain the vertical LC alignment and study the long alkyl group effects of chavicol unit on the LC alignment performances. The optical and electrical properties of the LC cells fabricated from the polymer films were also determined.

## 2. Materials and Methods

### 2.1. Materials

4-Chloromethylstyrene, chavicol, potassium carbonate, and a nematic LC, 4′–pentyl–4–biphenylcarbonitrile (5CB) (*n*_e_ = 1.7074, *n*_o_ = 1.5343, and Δε = 14.5, where *n*_e_, *n*_o_, and Δε represent extraordinary refractive indexes, ordinary refractive indexes, and dielectric anisotropy, respectively), were purchased from Aldrich Chemical Co. *N,N*′-Dimethylacetamide (DMAc) was dried using molecular sieves (4 Å). Tetrahydrofuran (THF) was dried by refluxing with benzophenone and sodium, followed by distillation. 4-Chloromethylstyrene was purified by column chromatography on silica gel using hexane as an eluent to eliminate any impurities and inhibitors (*tert*-butylcatechol and nitroparaffin). Poly(4-chloromethylstyrene) (PCMS) was synthesized via conventional free radical polymerization of the 4-chloromethylstyrene with 2,2′-azobisisobutyronitrile (AIBN) under a nitrogen atmosphere. AIBN (Junsei Chemical Co., Ltd., Tokyo, Japan) was used as an initiator. The AIBN was purified from crystallization using methanol. All other reagents and solvents were used as received.

### 2.2. Synthesis of Chavicol-Modified Polystyrene

PCHA#, chavicol–substituted polystyrene, where # represents the molar content (%) of the chavicol moiety in the side group, was synthesized by the following procedure. The synthesis of the chavicol–substituted polystyrene PCHA100 is presented as an example. Poly(4-chloromethylstyrene) (PCMS, 0.3 g, 1.97 mmol) was dissolved in *N,N*′–dimethylacetamide (DMAc, 50 mL). Chavicol (0.396 g, 2.95 mmol, 150 mol% compared with PCMS) and potassium carbonate (0.475 g, 2.99 mmol) were added to the PCMS solution. The solution mixture was magnetically stirred at 70 °C for 24 h under a nitrogen atmosphere. The synthesized solution mixture was poured into ethanol after cooling to room temperature to obtain a white precipitate. The precipitate was further purified over several reprecipitations from DMAc solution into ethanol and then washed with hot ethanol to eliminate potassium carbonate and any residue. The PCHA100 has obtained yields above 80% after drying in a vacuum overnight. The degree (%) of substitution of the chloromethyl to the chavicyl group was confirmed to be approximately 100% within experimental error.

PCHA100 ^1^H NMR (CDCl_3_): *δ* = 0.54–2.09 (m, 3H, –*CH_2_*–*CH*–Ph–), 3.15–3.36 (m, 2H, *CH_2_*–CH=CH_2_), 4.64–4.94 (s, 2H, –Ph–*CH_2_*–O–), 4.94–6.03 (d, 3H, CH_2_–*CH*=*CH*_2_), 6.20–7.23 (m, 8H, –CH_2_–CH–*PhH*–CH_2_–O–*PhH*–CH_2_).

Other polystyrene derivatives containing chavicol side groups were synthesized by a similar procedure used to prepare PCHA100 except for differing amounts of chavicol in the reaction. For example, PCHA80, PCHA60, PCHA40, and PCHA20 were prepared with chavicol of 0.211 g (1.57 mmol), 0.158 g (1.18 mmol), 0.106 g (0.79 mmol), and 0.053g (0.39 mmol), respectively, using slight excess amounts of potassium carbonate (120 mol% compared with chavicol).

### 2.3. Film Preparation and LC Cell Assembly

Solutions of each PCHA# in chloroform (1.0 wt.%) were prepared. These solutions were filtered using a poly(tetrafluoroethylene) (PTFE) membrane with a pore size of 0.45 μm. The thin films of the polymer were made by spin–coating (1st step at 700 rpm and 5 s, 2nd step at 1800 rpm and 90 s) onto glass substrates. LC cells were produced by assembling the polymer films using spacers with a thickness of 4.25 μm. The cells were filled with nematic liquid crystal, 5CB, and were sealed with epoxy glue. The alignments of the LC molecules onto the PCHA# films were observed with naked eyes and using the polarized optical microscopy (POM) after heat treatment at 50 °C above the clearing temperature (*T_c_*) of 5CB (*T_c_* = 35 °C) and cooling back to the room temperature.

### 2.4. Instrumentation

Proton nuclear magnetic resonance (^1^H NMR) measurement was carried out on a Bruker AVANCE spectrometer at 300 MHz. Differential scanning calorimetry (DSC) measurements were carried out on a TA instruments Q-10 (New Castle, DE, USA) at heating and cooling rates of 10 °C min^−1^ under a nitrogen atmosphere. The contact angles of distilled water and diiodomethane on the polymer films were measured using a Krüss DSA10 (KRÜSS scientific instruments Inc., Hamburg, Germany) contact angle analyzer equipped with drop shape analysis software. The surface energy value was calculated with Owens–Wendt’s equation:(1)$$\gamma_{sl}=\gamma_{s}+\gamma_{l}-2{\left(\gamma_{s}{}^{d}\gamma_{l}{}^{d}\right)}^{1/2}-2{\left(\gamma_{s}{}^{p}\gamma_{l}{}^{p}\right)}^{1/2}$$

Here, *γ_l_* is the surface energy of the liquid, *γ_sl_* is the interfacial energy of the solid/liquid interface, *γ_s_* is the surface energy of the solid, the superscripts *d* and *p* represent the dispersion and polar components of the surface energy. *γ_l_^d^* and *γ_l_^p^* are known for the test liquids, and *γ_s_^d^* and *γ_s_^p^* can be calculated from the measured static contact angles [55]. Polarized optical microscopy (POM) images of the LC cell were taken by an optical microscope (Nikon, ECLIPSE E600 POL, Tokyo, Japan) equipped with a polarizer and digital camera (Nikon, COOLPIX995, Tokyo, Japan).

## 3. Results

### 3.1. Synthesis and Characterization of Chavicol-Modified Plystyrene

The synthetic routes for chavicol–substituted polystyrene PCHA100 and copolymers PCHA80, PCHA60, PCHA40, and PCHA20, where # is the molar content (%) of chavicol side groups, are shown in Figure 1. The copolymers with different substitution ratios (%) were obtained by changing the chavicol amounts in the reaction. Almost 100% conversions from chloromethyl to chavicyl group were obtained when 150 mol% of chavicol was used at 70 °C for 24 h, as shown in the assignment of the respective proton peaks of the chavicol–substituted polystyrene PCHA100. The chemical compositions of monomeric units in the obtained polymers were confirmed by ^1^H NMR spectroscopy. ^1^H NMR spectrum of PCHA100 indicates the presence of protons from the phenyl ring of the styrene backbone (*δ* = 6.2–7.2 ppm [peak a]) (Figure 2). The proton peaks from chavicol side chains (*δ* = 3.2–3.4 [peak c], 4.9–6.0 ppm [peak d]) indicate the inclusion of chavicol moieties in the polymer. The chavicol content was calculated by analyzing the integration value of proton peaks of the chavicol (*δ* = 3.2–3.4 and 4.9–6.0 ppm, [peak c and d]) and chloromethyl (*δ* = 4.6–4.9 ppm, [peak b]). Similar integrations and calculations for PCHA80, PCHA60, PCHA40, and PCHA20 were performed and were typically within ± 10% of the predicted values of synthesis. These copolymers are soluble in medium–polarity solvents using low boiling points, such as chloroform. All samples have good solubility for various solvents to make thin-film materials for next-generation applications based on wet processes.

The thermal properties of these polymers, PCHA#, were examined using differential scanning calorimetry (DSC). All the polymers were amorphous because only one glass transition was observed from their DSC thermogram (Figure 3). As the molar content of chavicol side group increased from 20 to 100, the *T*_g_ value decreased from 98.4 °C for PCHA20 to 53.2 °C for PCHA100. The *T*_g_ value of the polystyrene derivatives decreased according to an increased molar content of the bulky side groups. It was ascribed to the increase of the sizeable steric volume in the polymer because polymers having larger steric volumes of substituents affect the flexibility of the polymer and lead to lower *T*_g_ values [56,57,58,59].

### 3.2. LC Alignment Behavior of the LC Cell Fabricated with Chavicol-Modified Polystyrene Film

Conoscopic polarized optical microscopy (POM) images of the LC cells fabricated with PCHA100 films onto glass substrates under the weight concentrations of the PCHA100 of 0.01, 0.05, 0.1, 0.5, 1.0, 1.5, and 2.0 wt.% are shown in Figure 4. At first, random planar alignment was observed in the PCHA100 weight ratios of less than 0.1 wt.% (Figure 4a–c). In the cases when the PCHA100 weight ratios were more than 0.5 wt.%, vertical alignments were observed as shown in the Maltese cross pattern of Figure 4d–g. The distinguishable differences in vertical LC alignment in the LC cells fabricated with PCHA100 of 1.0, 1.5, and 2.0 wt.% were not observed in Maltese cross pattern in the conoscopic images, as shown in Figure 4e–g. Therefore, 1.0 wt.% was selected as the coating concentration of the solution to fabricate LC cells made from these copolymers (PCHA#).

Photographic images of LC cells made from these copolymers (PCHA#) are shown in Figure 5. The LC cells fabricated using the PCHA# films with chavicol side group content of 20 mol% (PCHA20) shows a random planar LC alignment, while random planar and/or tilted LC alignment is observed for the LC cells fabricated using PCHA40 polymer films. The vertical LC alignment behavior was examined for the LC cells made from the polymer films having chavicol side group content larger than 60 mol% (PCHA60, PCHA80, and PCHA100). PCHA60, PCHA80, and PCHA100 films were maintained for at least several months after fabricating the LC cells from these polymers. Therefore, as the molar content of chavicol side group in PCHA# increases, the vertical LC alignment also increases.

As shown in Figure 6, the LC alignment behaviors on PCHA# films were investigated by observing the POM images. Random planar LC alignment behaviors were observed for the LC cells fabricated from the PCMS film (figure not shown). When the molar fraction of the chavicol containing monomeric unit in the PCHA# was 20 mol%, the LC cells fabricated from the PCHA# film exhibited planar LC alignment by the conoscopic POM images. On the other hand, the partial random planar and the partial vertical LC alignment was observed simultaneously in the conoscopic POM image of the LC cell fabricated with the PCHA40 film. However, the uniform vertical LC alignment of the LC cells made from the polymer films (PCHA60, PCHA80, and PCHA100) was observed by Maltese cross pattern in conoscopic POM images.

### 3.3. Surface Properties of Chavicol-Modified Polystyrene Films

Based on the LC alignment results, this study obtained a consistent trend that the polymers with higher molar content of chavicol side groups prefer vertical LC alignment. It is known that the vertical alignment behavior is related to either or both the low surface energy on the alignment layer surfaces and the large steric repulsions between LCs and the alignment layers [60,61]. For instance, polyimide derivatives incorporate nonpolar and bulky side groups featuring vertical alignment behavior such as pentylcyclohexylbenzene [60] and 4-(*n*-octyloxy)phenyloxy [61]. Therefore, this research demonstrated the LC alignment performances on the PCHA# films via surface characterization techniques such as surface energy value measurements. Surface energy values of the polymer films calculated from the measured static contact angles of water and diiodomethane are shown in Figure 7 and Table 1. The total surface energy was calculated with the Owens–Wendt’s equation, and this is a summation of the polar and dispersion contributions. The critical surface energy value of the polymer films to give vertical LC alignment performances was also obtained. The vertical LC alignment was exhibited from the PCHA60, PCHA80, and PCHA100 films. The total surface energy values of PCHA# films following the molar content of the chavicol moiety in the side groups increased to 44.5, 45.4, 46.7, 48.2, and 48.6 mJ m^−2^. The dispersion surface energy values per the molar content of the chavicol moiety also increased, to 40.1, 43.3, 45.4, 47.2, and 47.7 mJ m^−2^. On the other hand, the polar surface energy values of PCHA# films according to the molar content of the chavicol moiety in the side groups decreased to 4.4, 2.1, 1.3, 1.0, and 0.9 mJ m^−2^. This study observed the correlation between the vertical LC alignment and the polar surface energy value of the copolymer films. Other researchers have reported that polar surface energy of polymer film can influence the LC alignment performances [62,63,64]. Chavicol has a nonpolar and bulky group, such as an allyl group attached to the phenyl group in the *para* position, and not only a phenyl group. Thus, the increase in the molar content of chavicol leads to a decrease in the polar surface energy value, in which vertical alignment can be formed. Therefore, it is reasonable to suggest that the vertical LC alignment performances of PCHA100, PCHA80, and PCHA60 are induced by the increased steric repulsions between the LCs and the polymer surfaces. These repulsions result from the nonpolar and bulky chavicol moieties into the side group of the polystyrene and the low polar surface energy emerging from the unique chemical structures.

### 3.4. Reliability and Electro-Optical Performance of the LC Cells Fabricated with Chavicol-Modified Polystyrene Films

The reliabilities of LC cells made from the polymer films were confirmed by a stability test of the LC alignment behavior under harsh conditions and ultraviolet (UV) irradiation. The UV stability of the LC cells fabricated with the PCHA100 film was demonstrated from the POM image after UV irradiation at 0.1, 0.5, 1, and 5 J/cm^2^. No distinguishable difference of the Maltese cross pattern in the conoscopic POM images on PCHA100 film having vertical LC aligning ability could be observed (Figure 8), indicating that the vertical LC aligning ability of the PCHA100 LC cell was found to be maintained even at high UV energy.

Therefore, using plant-based and renewable resources, PCHA# can be considered next-generation LC alignment layers for eco–friendly 4th industrial revolution applications.

## 4. Conclusions

A series of plant-based chavicol-modified polystyrene (PCHA#) was synthesized in order to investigate the alignment behavior of LC molecules as an anisotropic material on polymer films. The LC cells fabricated with the polymer films having chavicol units larger than 60 mol% (PCHA60, PCHA80, and PCHA100) showed vertical LC alignment. However, the LC cells fabricated with PCHA# films having chavicol smaller than 40 mol% exhibited random planar LC alignment behavior. The vertical LC alignment was exhibited by the steric repulsions between the LCs and the polymer surface because of a nonpolar and bulky chavicol moiety into the side chain. There is a good correlation between the vertical alignment behavior of LC molecules and the polar surface energy values of these copolymer films having smaller than approximately 1.3 mJ/m^2^, induced by the long alkyl groups. These results show the basic idea for the advanced research of eco–friendly and sustainable LC alignment layers based on renewable and phytochemical-based chavicol containing polymer films.

## Figures and Tables

**Figure 1 polymers-13-00819-f001:**
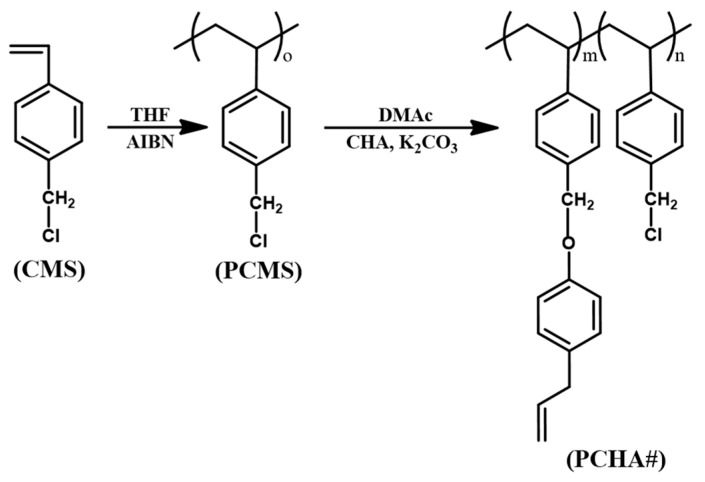
Synthetic route of chavicol-substituted polystyrene films (PCHA#), where # indicates the mole percent of chavicol containing monomeric units in the polymer.

**Figure 2 polymers-13-00819-f002:**
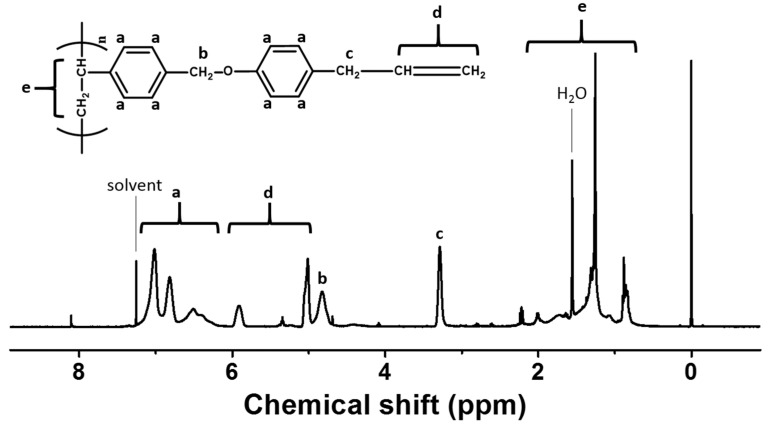
Proton nuclear magnetic resonance (^1^H NMR) spectrum of PCHA100.

**Figure 3 polymers-13-00819-f003:**
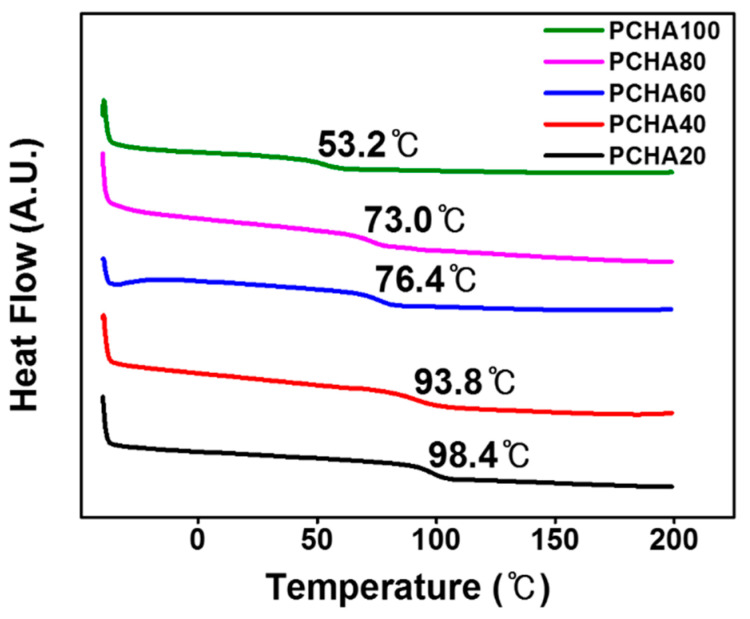
Differential scanning calorimetry (DSC) thermogram of PCHA#.

**Figure 4 polymers-13-00819-f004:**
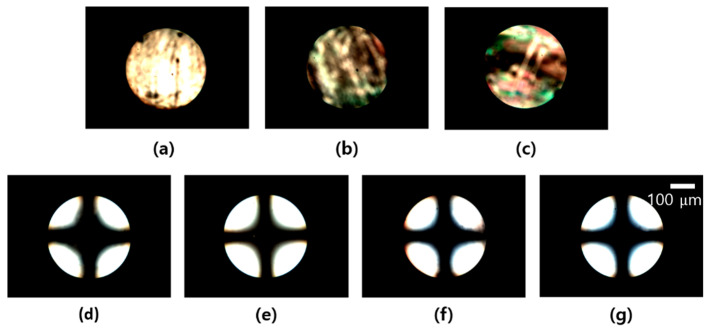
Conoscopic polarized optical microscopy (POM) images of the cells fabricated with PCHA100 films under the following a weight ratio of the PCHA100; 0.01 (**a**), 0.05 (**b**), 0.1 (**c**), 0.5 (**d**), 1.0 (**e**), 1.5 (**f**), and 2.0 wt.% (**g**); scale bar: 100 μm, all scale bars are the same.

**Figure 5 polymers-13-00819-f005:**
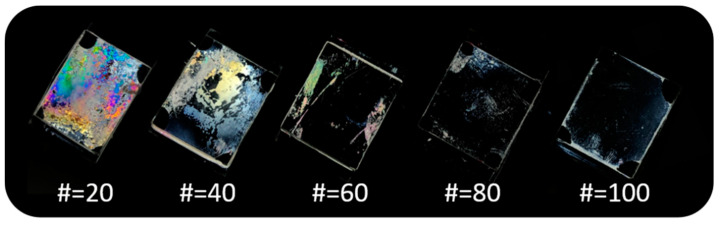
Photograph images of the LC cells fabricated with PCHA# films according to the molar content of chavicol moiety.

**Figure 6 polymers-13-00819-f006:**
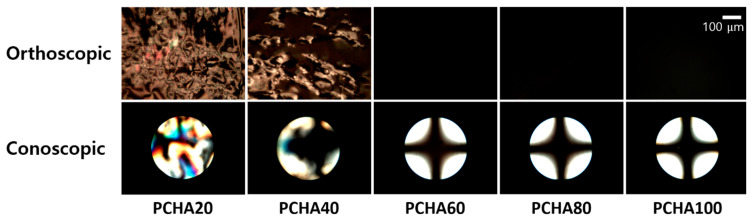
Orthoscopic and Conoscopic polarized optical microscopy (POM) images of the LC cells made from PCHA# (PCHA20, PCHA40, PCHA60, PCHA80, and PCHA100) films; scale bar: 100 μm, all scale bars are the same.

**Figure 7 polymers-13-00819-f007:**
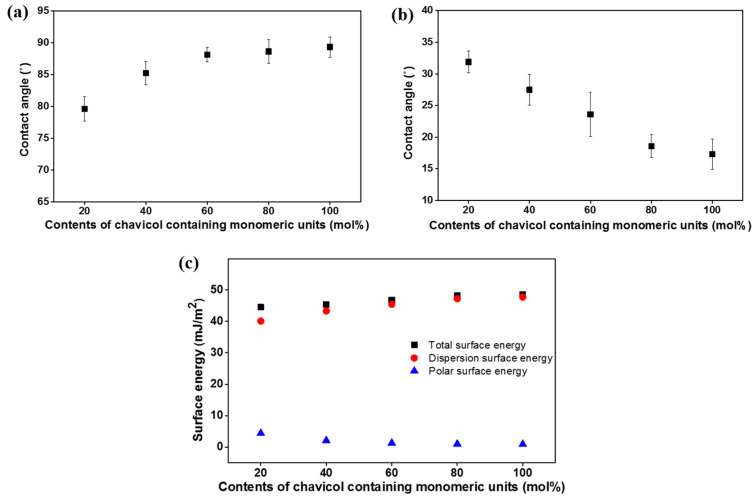
(**a**) Water, (**b**) diiodomethane contact angle and (**c**) surface energy values of PCHA# films according to the molar contents of the chavicol moiety in the side groups.

**Figure 8 polymers-13-00819-f008:**
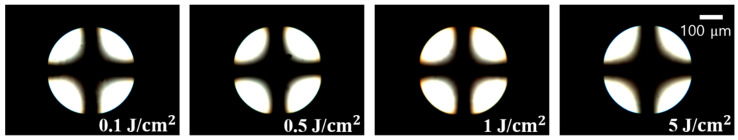
Conoscopic polarized optical microscopy (POM) images of the LC cells fabricated with PCHA100 films, after UV treatment at 0.1, 0.5, 1, and 5 J/cm^2^, respectively; scale bar: 100 μm, all scale bars are the same.

**Table 1 polymers-13-00819-t001:** Surface energy values and LC alignment properties.

Polymer Designation	Contact Angle (°) *^a^*		Surface Eergy (mJ m^−2^) *^b^*			LC Aligning Ability *^c^*
	Water	Diiodomethane	Polar	Dispersion	Total	
PCHA20	79.7	31.6	4.4	40.1	44.5	X
PCHA40	83.9	29.1	2.1	43.3	45.4	X
PCHA60	87.9	20.8	1.3	45.4	46.7	O
PCHA80	88.5	20.7	1.0	47.2	48.2	O
PCHA100	88.7	18.8	0.9	47.7	48.6	O

*^a^* Measured from static contact angles. ^*b*^ Calculated from Owens–Wendt’s equation. *^c^* Circle (O) and cross (X) indicate polymer film have vertical and random planar LC aligning ability, respectively.

## Data Availability

The data presented in this study are available on request from the corresponding author.

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
