# Peer review of "Vertical Alignment of Liquid Crystals on Comb-Like Renewable Chavicol-Modified Polystyrene"

_polymers, 2021, doi:10.3390/polym13050819_

Round 1
Reviewer 1 Report
see the uploaded report

Author Response
Dear Editor at Polymers
We gratefully appreciate your kind reviewing and considering for publication in “Polymers”. We are submitting a revised manuscript (polymers-1132533) entitled “Vertical Alignment of Liquid Crystals on Comb-Like Renewable Chavicol-modified Polystyrene”.
We carefully read the reviewer’s comments and your e-mail. Reviewers gave us helpful comments for our manuscript. We think the reviewer’s opinion and suggestion is fairly reasonable. Therefore, we revised our manuscript taking the reviewer’s comments into consideration as follows. As you and the Reviewer suggested we modified some parts of the manuscript and the changes are shown as yellow texts. These changes are listed as follows:
Referee’s comments:
Referee: 1
Comments:
In my view the manuscript can be recommended for a publication. The minor points the authors may want to address are listed below.
- When talking about the LC alignment methods in the Introduction it seems useful to mention the Photoalignment technique (e.g. book Vladimir G. Chigrinov, Vladimir M. Kozenkov and Hoi‐Sing Kwok “Photoalignment of Liquid Crystalline Materials: Physics and Applications”, John Wiley & Sons,, 2008; A.G. Dyadyusha, A. Khizhnyak, T Marusii, Y. Reznikov, V.Yu. Reshetnyak O. Yaroshchuk, W. Park, S. Kwon, H. Shin and D. Kang An oblique orientation of nematic liquid crystals on a photosensitive aligning polymer. Mol. Cryst .Liq. Cryst, 263, 399-414. (1995));
Answer:
We deeply thanks for comments suggested by Reviewer. We strongly believe that Reviewer’s suggestion might be very helpful to increase the understanding of our manuscript by readers. As the Reviewer suggested we added the several works for photoalignment techniques into References as number of 21–23. Added literatures are below.
21. Chigrinov, V.G.; Kozenkov, V.M.; Kwok, H. Photoalignment of Liquid Crystalline Materials: Physics and Applications.; John Wiley & Sons, UK, 2008.
22. Dyadyusha, A.; Khizhnyak, A.; Marusii, T.; Reznikov, Y.; Yaroshchuk, O.; Reshetnyak, V.; Park, W.; Kwon, S.; Shin, H.; Kang, D. An oblique orientation of nematic liquid crystals on a photosensitive aligning polymer. Mol. Cryst. Liq. Cryst. 1995, 263, 399–413.
23. Yin, K.; Xiong, J.; He, Z.; Wu, S. Patterning liquid-crystal alignment for ultrathin flat optics. ACS Omega 2020, 5, 31485–31489.
- The authors say: “At a molar content of 40 mol%, the LC cells fabricated with the PCHA# film exhibited a random tilted LC alignment in the conoscopic POM images.” How big is the tilt? Is it the same over the sample or varies from point to point? The authors might want to use the term degenerate planar alignment;
Answer:
We deeply thanks for comments suggested by Reviewer and we revised text as “On the other hand, the partial random planar and the partial vertical LC alignment was observed simultaneously in the conoscopic POM image of the LC cell fabricated with the PCHA40 film.” We hope that the Reviewer satisfy our correction in the revised manuscript.
- Further on the authors say: ”Therefore, it is reasonable to demonstrate that the vertical LC alignment performances of PCHA100, 233 PCHA80, and PCHA60 are induced by the increased steric repulsions between the LCs 234 and the polymer surfaces.” I’d use “ … to make a guess or … to suggest;
Answer:
We deeply thanks for comments suggested by Reviewer. As the Reviewer kindly suggested we revised as “Therefore, it is reasonable to suggest that the vertical LC alignment performances of PCHA100, PCHA80, and PCHA60 are induced by the increased steric repulsions between the LCs and the polymer surfaces.”
- “The reliabilities of LC cells made from the polymer films were confirmed by a stability test of the LC alignment behavior under harsh conditions and ultraviolet (UV) irradiation.” Did the authors check if the alignment is stable under the cell heating above the LC clearing point (let’s say up to 70C) and then cooling back?
Answer:
We are much obliged for comment suggested by Reviewer and we added the explanation about experimental process in Materials and Methods section (page 3, lines 115–118) as “The alignments of the LC molecules onto the PCHA# films were observed with naked eyes and using the polarized optical microscopy (POM) after heat treatment at 50 °C above the clearing temperature (Tc) of 5CB (Tc = 35 °C) and cooling back to the room temperature.” We hope that the Reviewer satisfy our correction in the revised manuscript.
Referee: 2
Comments:
In this manuscript, the authors synthesize the comb-like PCHA# for VA mode alignment and study the effects on LC alignment. The optical and electrical properties of the fabricated cell are also determined and discussed. I personally recommend the manuscript to be published in Polymers if the following comments could be taken into consideration.
-In Figure 2, could the authors explain what the peaks on both sides of peak f are? The peaks are sorted from (a) to (f), but (e) is skipped in the middle.
Answer:
We deeply thanks for comments suggested by Reviewer. We revised Figure 2 and the presentation of 1H-NMR data including chemical shift of peaks and area in Materials and Methods section as below.
Figure 2. Proton nuclear magnetic resonance (1H NMR) spectrum of PCHA100.
PCHA100 1H NMR (CDCl3): δ = 0.54–2.09 (m, 3H, –CH2–CH–Ph–), 3.15–3.36 (m, 2H, CH2–CH=CH2), 4.64–4.94 (s, 2H, –Ph–CH2–O–), 4.94–6.03 (d, 3H, CH2–CH=CH2), 6.20–7.23 (m, 8H, –CH2–CH–PhH–CH2–O–PhH–CH2).
-In Figure 4, the alignment is observed from 0.5 to 1 wt%. However, we can still see the obvious difference between (d) and (e). Would it be better or worse if the weight percent is keep increasing, like 1.5 wt% and 2 wt%?
Answer:
We are much obliged for comment suggested by Reviewer. As the Reviewer kindly suggested we observed the conoscopic POM images of the LC cells fabricated with PCHA100 films onto glass substrates under the following a weight concentration of the 1.5 wt.% and 2.0 wt.%. We added these results into Figure 4 and added text in Results section (page 5, lines 175–178) as “The distinguishable differences in vertical LC alignment in the LC cells fabricated with PCHA100 of 1.0, 1.5, and 2.0 wt.% were not observed in Maltese cross pattern in the conoscopic images, as shown in Figures 4(e)–(g).” We hope that Reviewer will satisfy the answer about this question.
Figure 4. Conoscopic polarized optical microscopy (POM) images of the cells fabricated with PCHA100 films under the following a weight ratio of the PCHA100; 0.01, 0.05, 0.1, 0.5, 1.0, 1.5, and 2.0 wt.%.
-In order to provide the readers a more comprehensive understanding of the research background, I suggest adding the following works in the revision: For example, LC photoalignment (Yin, J. Xiong, Z. He, and S. T. Wu, “Patterning Liquid Crystal Alignment for Ultra-Thin Flat Optics,” ACS Omega 5, 31485-31489, 2020); LC devices and applications: (K. Yin, Z. He, and S. T. Wu, “Reflective polarization volume lens with small f-number and large diffraction angle,” Adv. Opt. Mater. 8, 2000170, 2020).
Answer:
We deeply thanks for comments suggested by Reviewer. We strongly believe that Reviewer’s suggestion might be very helpful to increase the understanding of our manuscript by readers. As the Reviewer suggested we added the several works for photoalignment techniques as number of 21–23 and LC devices and applications as number of 7 in the References. Added literatures are below.
7. Yin, K.; He, Z.; Wu, S. Reflective polarization volume lens with small f‐number and large diffraction angle. Adv. Opt. Mater. 2020, 8, 2000170.
21. Chigrinov, V.G.; Kozenkov, V.M.; Kwok, H. Photoalignment of Liquid Crystalline Materials: Physics and Applications.; John Wiley & Sons, UK, 2008.
22. Dyadyusha, A.; Khizhnyak, A.; Marusii, T.; Reznikov, Y.; Yaroshchuk, O.; Reshetnyak, V.; Park, W.; Kwon, S.; Shin, H.; Kang, D. An oblique orientation of nematic liquid crystals on a photosensitive aligning polymer. Mol. Cryst. Liq. Cryst. 1995, 263, 399–413.
23. Yin, K.; Xiong, J.; He, Z.; Wu, S. Patterning liquid-crystal alignment for ultrathin flat optics. ACS Omega 2020, 5, 31485–31489.
We believe that now we answered all of the comments pointed out by the reviewers. I hope that now this paper is publishable in “Polymers”, one of the top journals in polymer science area.
We also believe that this paper is also suitable for publication in “Polymers” from the following reasons.
- We synthesized phytochemical-based and renewable chavicol-modified polystyrene (PCHA#, # = 20, 40, 60, 80, and 100, where # is the molar content of chavicol moiety in the side group) to investigate LC alignment behavior on the surface of polymer films for the first time. The optical properties of the polymer films were also investigated.
- Polyimide derivatives are most widely used as conventional LC alignment layers. However, baking processes needed curing temperature higher than 200 oC to produce polyimide films cannot be used in practical for several applications. We believe that renewable chavicol-substituted polystyrene can produce an eco-friendly and sustainable LC alignment layer for next generation applications without baking processes.
- Since we could vary the mole percent of the chavicol groups, the effect of chavicol moieties in the side chain on their LC alignment behaviors could be studied very systematically. The surface characterization such as surface energy measurement was carried out in order to investigate the effect of the wettability on the LC alignment properties of the polymer film. The surface energy value on the polymer film was correlated with the LC alignment behavior. For example, the vertical alignment was observed when the polar surface energy values of the polymer were smaller than about 1.3 mJ/m2 induced by the nonpolar and long carbon groups.
- We believe that this can contribute the basic idea for the development of LC alignment layers based on renewable chavicol resource containing polymer made from simple polymer reaction having good solubility in many organic solvents and their films having low process temperature.
In view of these achievements, we believe that our work represents a timely methodological advance and breakthrough in the field of polymer science and thus is appropriate for a journal with the scope and wider readership of “Polymers”.
I hereby certify that this manuscript consists of original, unpublished work which is not under consideration for publication elsewhere.
We are excited to share our manuscript with you and look forward to hearing good news from you soon.
Thank you very much for your time and consideration for the process.
Sincerely (on behalf of all authors),
Prof. Hyo Kang
Associate Professor
Department of Chemical Engineering (BK-21 Four Graduate Program)
Dong-A University
Busan 604-714, Republic of Korea
Tel: +82 51 200 7720
Fax: +82 51 200 7728
E-mail hkang@dau.ac.kr

Reviewer 2 Report
Please find the review report as attached.

Author Response

(The authors gave the same response as above.)
